# Functional Response and Intraspecific Competition of Three Ladybird Species Feeding on Aphids on Goji Berry Plants in Laboratory and Semi-Field Conditions

**DOI:** 10.3390/insects14110853

**Published:** 2023-10-31

**Authors:** Pengxiang Wu, Jia He, Huan Dong, Runzhi Zhang

**Affiliations:** 1Key Laboratory of Zoological Systematics and Evolution, Institute of Zoology, Chinese Academy of Sciences, Beijing 100101, China; wupengxiang@ioz.ac.cn; 2Institute of Plant Protection, Academy of Ningxia Agriculture and Forestry Science, Yinchuan 750002, China; hejiayc@126.com; 3Hami Plant Quarantine Workstation, Department of Agriculture and Rural Affairs of Xinjiang Uygur Autonomous Region, Hami City 839099, China; tianhappy1106@126.com; 4College of Life Science, University of Chinese Academy of Sciences, Beijing 100049, China

**Keywords:** functional response, intraspecific competition, practical application, biological control

## Abstract

**Simple Summary:**

The aphid *Aphis gossypii* Glover is a serious pest that inflicts severe damage upon goji berry plants in China. The current prevailing approach to pest prevention involves the utilization of chemical insecticides, which presents potential risks to both human health and the environment. The improper use of pesticides leads to the emergence of drug-resistant pests, thereby contributing to the resurgence of rampant pest populations. Therefore, the conservation and management of predators in agricultural ecosystems should receive increased attention, given their crucial role. Ladybirds have previously been identified as the primary predators of aphid species. However, their effectiveness as biological agents against aphids on goji berry plants remains uncertain. We found that the potential of ladybirds in controlling the aphid populations is promising, thus possibly making a contribution to environmental protection. By conducting an analysis of functional responses, intraspecific competition, and a semi-field study, we have determined that *H. axyridis* and *C. septempunctata* exhibit greater potential as biocontrol agents against aphids in comparison to *H. variegata*. Notably, future field studies will play a pivotal role in ensuring the effective implementation of a biological control program.

**Abstract:**

The aphid, *Aphis gossypii* Glover, is identified as a significant pest that causes severe damage to goji berries in China. To analyze the ladybird consumption of aphids, the functional responses of three ladybird species, *Harmonia axyridis*, *Coccinella septempunctata*, and *Hippodamia variegata*, and intraspecific competition among ladybird individuals were evaluated under laboratory conditions. Moreover, the practical impact of ladybirds on aphid population reduction was investigated in semi-field conditions. We found that all adult ladybirds of the three species exhibited a type II functional response toward aphids. According to Holling’s disc equation, *H. axyridis* exhibited the highest searching efficiency (*a* = 0.79), while *C. septempunctata* had the shortest handling time (*T*_*h*_ = 5.07 min) among the three ladybird species studied. Additionally, intraspecific competition had a greater impact on *H. variegata* (*m* = 0.41) compared to the other two ladybird species. The semi-field study demonstrated that *H. axyridis* (83.9% reduction) and *C. septempunctata* (78.7% reduction) exhibited higher efficacy in reducing aphid populations compared to *H. variegata* (27.3% reduction). This study suggests that *H. axyridis* and *C. septempunctata* exhibit potential as effective biological control agents against aphids on goji berry plants and highlights the importance of considering intraspecific competition. However, the results obtained from laboratory and semi-field studies cannot be directly extrapolated to field conditions due to the simplification of these experimental systems. Future field studies are crucial in ensuring the effective implementation of a biological control program.

## 1. Introduction

The goji berry, *Lycium barbarum* L. (Solanaceae: Lycium Linn.), is a prominent plant endemic to China. The efficacy of goji berry has been extensively investigated and empirically validated in terms of its capacity to augment sleep quality and provide a diverse array of supplementary health advantages [1,2,3]. Because of its flourishing leaf growth, sweet fruit, and nutrient-rich composition, goji berries are highly susceptible to pest infestations and disease outbreaks. The vegetative and reproductive growth periods of goji berry exhibit a high overlap. The duration of the young fruit stage in goji berry is extended, whereas the periods of fruit expansion and maturation are comparatively brief. Pest control in goji berry cultivation poses a formidable challenge [4,5]. Furthermore, inadequate chemical control not only seldom attains the desired level of efficacy, but it also readily gives rise to significant concerns such as pesticide residue and environmental pollution [6,7,8,9,10]. The frequent occurrence of pests, influenced by recent climatic factors, has led to significant losses in both the quantity and quality of goji berries. Particularly, the presence of aphid populations is frequently observed on goji berry plants [11].

The majority of this aphid population has been identified as *Aphis gossypii* Glover (Homoptera: Aphididae), which is commonly observed in goji berry-growing regions, exhibiting a broad geographical distribution [12]. The pest exhibits clustering behavior and inflicts damage on newly emerged shoots, stems, and leaf surfaces through its sap-sucking activity. This results in the stunted growth of new shoots and the development of narrow brown scorched margins on leaves followed by withering and aberrant fruit formation [13,14]. The leaf surfaces become coated with honeydew, which significantly impairs photosynthetic efficiency, disrupts reproductive processes, and inhibits plant blooming [15].

The majority of previous studies pertaining to the aphid have primarily focused on its occurrence and population dynamics, as well as the utilization of parasitoids and their host acceptance behavior [16,17]. Additionally, research has been conducted on toxicity tests in relation to the aphid, along with strategies aimed at enhancing insect resistance [18]. Regarding its predators, functional responses of ladybird larvae to *A. gossypii* have been investigated both in laboratory settings and in melon fields using molecular gut content analysis to assess predation [19,20,21,22,23]. However, there is currently limited research on the potential of adult ladybirds for controlling aphids on goji berry plants, and our study aimed to establish a theoretical foundation for their biological control. In addition, the current predominant approach to pest prevention involves the utilization of chemical pesticides, which pose potential risks to both human health and the environment [24,25,26]. The misuse of pesticides also results in the emergence of drug resistance, thereby contributing to the resurgence of rampant pest populations [7,8,27,28]. Therefore, the conservation and management of predators in agricultural ecosystems warrants greater attention. Ladybirds have been previously reported as the primary predators of aphid species [29]; however, their effectiveness as biological agents against this aphid species remains uncertain. The potential of ladybirds in controlling invasive pest populations is highly promising, suggesting that they could make a contribution to environmental protection.

In order to assess the ladybird consumption of aphids on goji berry plants, we selected three common predatory ladybird species: *Harmonia axyridis* Pallas, *Coccinella septempunctata* Linnaeus, and *Hippodamia variegata* Goeze. We aimed to address three key questions: (1) the functional response of adult ladybirds preying on aphids, (2) the intraspecific competition among adult ladybirds, and (3) the practical effect of ladybirds in reducing aphid populations in a semi-field setting.

## 2. Materials and Methods

### 2.1. Insect Rearing

The colonies of the ladybird species *H. axyridis*, *C. septempunctata*, and *H. variegata*, as well as the aphid *A. gossypii*, were obtained from the Dadi eco-cultivation bases of goji berry in Zhongning County, Ningxia Province, China. The aphids were reared and propagated on goji berry leaves under controlled conditions of 16 L: 8 D, 25 °C and 40–60% RH. Approximately 20 larvae or adults belonging to the same ladybird species were reared per plastic container (16 cm × 22 cm × 8 cm). The ladybird colonies selected were transferred into Petri dishes (9 cm diameter) containing goji berry leaves infested with aphids. The neonates of the ladybirds were promptly separated upon hatching to mitigate sibling cannibalism. To ensure a thriving ladybird population, aphids were provided at 12 h intervals. In the experiments, newly emerged (<12 h) adult ladybirds were initially subjected to a 24 h period of fasting, followed by placement in Petri dishes containing recently hatched (<12 h) adult aphids. The dishes were preserved under the same climatic conditions as previously described. Adult aphids remained unchanged throughout the experiments, while nymphs originating from the adult aphids were gently removed every 2 h using gentle brushes.

### 2.2. Functional Response 

To investigate the functional response of adult ladybirds to aphids, we conducted an experiment employing a fully factorial design in which we independently manipulated three ladybird species (*H. axyridis*, *C. septempunctata*, or *H. variegata*) and five levels of initial aphid density (100, 150, 200, 250, or 300 individuals). The appropriate number of aphids was placed on a moistened paper disc in a 9 cm diameter Petri dish to establish our experimental treatments. Subsequently, a single adult ladybird was introduced into each Petri dish, followed by an examination of aphid consumption by the ladybirds using a binocular microscope after 24 h. The control treatments were implemented to account for the inherent mortality of adult aphids and the fecundity of newborn nymphs, thereby adjusting for ladybird predation on the aphids. Each treatment was replicated five times. 

### 2.3. Intraspecific Competition 

To evaluate the influence of intraspecific competition on aphid consumption by adult ladybirds, three ladybird species (*H. axyridis*, *C. septempunctata*, or *H. variegata*) were independently manipulated. In a Petri dish, 100, 200, 300, 400 or 500 aphids were provided to 1, 2, 3, 4 or 5 ladybirds, respectively (5 treatments), maintaining a prey-to-predator ratio of 100. The intensity of intraspecific competition among ladybirds for space exhibited a positive correlation with the abundance of ladybirds in a Petri dish. The parameter assessing intraspecific competition was determined by quantifying the number of surviving aphids after a 24 h period. Each treatment was replicated 5 times.

### 2.4. Semi-Field Study

The practical efficacy of the biological control program against aphids was evaluated through a semi-field study. The optimal aphid-to-predator ratio was initially estimated based on the theoretical maximum consumption derived from the functional response model mentioned above. The initial aphid abundance was recorded on a goji berry plant within a mesh cage (60 mesh, 1 m diameter, 1.5 m height). This study involved the collection of a total of 60 leaves, with three layers and four directions from the tree. Subsequently, the average number of aphids per leaf was calculated and multiplied by the total number of leaves on the plant to determine the aphid abundance. The adult ladybirds (*H. axyridis*, *C. septempunctata*, or *H. variegata*) were introduced into the cage in an appropriate number, maintaining the optimal prey-to-predator ratio. The aphid abundance within the mesh cage was reassessed after a 30-day period. Control treatments were implemented without the presence of ladybirds to account for natural fluctuations in the aphid populations and to accurately assess the efficacy of the biological control program. All treatments were each replicated 3 times for a total of 12 plants.

### 2.5. Statistical Analysis

#### 2.5.1. Functional Response

The functional responses of ladybirds feeding on aphids on goji berry plants were determined through a two-stage analysis [30]. The first step involved conducting cubic logistic regression analysis to determine the shape (type II or type III) of the functional responses by examining the proportion of aphids consumed as a function of their initial density:(1)$$\frac{N_{a}}{N_{0}}=\frac{\exp(P_{0}+P_{1}N_{0}+P_{2}{N_{0}}^{2}+P_{3}{N_{0}}^{3})}{1+\exp(P_{0}+P_{1}N_{0}+P_{2}{N_{0}}^{2}+P_{3}{N_{0}}^{3})}$$

The equation is defined as follows: *N*_a_ represents the number of aphids consumed, while *N*_0_ denotes the initial aphid density. *P*_0_, *P*_1_, *P*_2_ and *P*_3_ correspond *to* the intercept, linear, quadratic, and cubic parameters, respectively. The presence of negative or positive linear parameters (*P*_1_) indicates a type II or type III functional response, respectively. If the parameters of a cubic equation are not statistically significant, it is recommended to simplify the model by removing the quadratic and cubic parameters from Equation (1) and to retest the remaining parameters. The logistic regression analysis revealed that our data exhibited a type II functional response in each case; thus, subsequent analyses were exclusively focused on the type II functional response. Holling’s disc (Equation (2)) [31] was employed to depict the correlation between the consumption of aphids (*N*_a_) and their initial density (*N*_0_):(2)$$N_{a}=\frac{a{TN}_{0}}{1+aT_{h}N_{0}}\;$$
where *N*_a_ and *N*_0_ are defined in Equation (1), *T* represents the total duration, which is 24 h in this case, *a* denotes the ladybird’s searching efficiency (i.e., the area covered per unit time), and *T*_*h*_ signifies the handling time required for processing one prey [32]. The parameters *a* and *T*_*h*_ were estimated using a nonlinear regression procedure (NLR) based on the Levenberg–Marquardt method. The initial values of *a* and *T*_*h*_ required by the NLR procedure were determined through the linear regression analysis of 1/*N*_a_ against 1/*N*_0_. The resulting *y* intercept serves as the initial estimation of *T*_*h*_, while the reciprocal of the regression parameter provides an estimate of *a* [33]. The theoretical maximum consumption by ladybirds (*N*_max_) can be obtained as *N*_0_ approaches infinity.

#### 2.5.2. Intraspecific Competition

This experiment was conducted to quantify the parameter of intraspecific competition among ladybird individuals during predation events. The estimation of the parameters for intraspecific competition was conducted through nonlinear regression analysis by fitting Equation (3) [34]:*E* = *qP*^−*m*^(3)

The equation is defined as follows: *E* represents the per capita consumption rate of ladybirds, *P* denotes the ladybird abundance, *m* signifies the parameter of intraspecific competition, and *q* indicates the maximum per capita consumption rate. The model was established for conducting regression analysis on a power function curve.

The descriptive statistics are presented as the mean values accompanied by their corresponding standard errors of the mean. All data were checked for a normal distribution and homoscedasticity. The differences between the natural mortality rate and zero were assessed using a one sample *t*-test; significance was determined at *p* < 0.05. The remaining data were subjected to one-way ANOVA analysis, followed by the Tukey HSD test for significance at a 5% level of statistical significance. The statistical analysis was conducted using SPSS 20.0 software (IBM, Armonk, NY, USA), while the regression analyses were performed utilizing SigmaPlot 13.0 software (Systat Software Inc., San Jose, CA, USA).

## 3. Results

### 3.1. Functional Response

Regardless of the aphid density, the natural mortality rates of the aphids did not exhibit a significant deviation from zero (*t*-test, *p* > 0.05) due to the limited production of newborn nymphs during the experiments. Consequently, both the mortality rates and the number of newborn nymphs among aphids remained inconsequential throughout the experimental trials. The parameter estimates derived from the logistic model (Equation (1)) for the proportion of aphids consumed by ladybirds within a 24 h period, in relation to the aphid density, are presented in Table 1. The estimated values of the linear parameter *P*_1_ were found to be significantly negative for all three species of ladybirds, indicating a type II response of these three ladybird species according to the logistic model analysis. 

The functional response data for aphid consumption by adult ladybirds over a 24 h period exhibited a strong fit to Holling’s disc model (Equation (2)), as evidenced by the results presented in Table 2. This confirms the presence of a type II response across all three ladybird species. The consumption of aphids gradually increased with an increase in the initial aphid densities (Figure 1A–C), and both *H. axyridis* (90.6 ± 4.4) and *C. septempunctata* (89.6 ± 4.5) exhibited significantly higher mean aphid consumption compared to *H. variegata* (56.6 ± 3.1) (*F*_2,72_ = 22.874, *p* < 0.001; Figure 1D). The numerical values of the parameters for search efficiency (*a*) and handling time (*T*_*h*_) exhibited this relationship, with asymptotic 95% confidence intervals excluding zero, across all three ladybird species. We found that *H. axyridis* (*a* = 78.6%) displayed a significantly greater rate of spatial coverage per unit time than both *C. septempunctata* (*a* = 68.8%) and *H. variegata* (*a* = 43.1%), whereas *C. septempunctata* (*T*_*h*_ = 5.07 min) exhibited a notably shorter duration for processing each individual aphid when compared with *H. axyridis* (*T*_*h*_ = 6.21 min) and *H. variegata* (*T*_*h*_ = 7.97 min). Moreover, the theoretical maximum aphid consumption by *C. septempunctata* (*N*_max_ = 284.2) exceeded that of *H. axyridis* (*N*_max_ = 231.8) and *H. variegata* (*N*_max_ = 180.6).

### 3.2. Intraspecific Competition

As the population sizes of the introduced aphids and ladybirds increased, there was a gradual augmentation in the total consumption of aphids in a Petri dish while maintaining a ratio of 100 aphids to ladybirds. However, irrespective of the ladybird species, the consumption rate of the ladybirds declined with increasing densities of both aphids and ladybirds due to intraspecific competition arising from spatial limitations (Figure 2A–C). The overall consumption rate by ladybirds demonstrated a significant decline with increasing densities of aphids and ladybirds (*F*_4,70_ = 5.368, *p* = 0.001; Figure 2D), with 48.8% ± 3.4%, 42.3% ± 3.8%, 35.0% ± 3.8%, 33.8% ± 4.5%, and 26.4% ± 2.8% when the density of ladybirds varied from one to five, respectively. Besides, the mean consumption rates of *H. axyridis* (45.5% ± 3.1%) and *C. septempunctata* (41.5% ± 2.7%) were higher than that of *H. variegata* (24.7% ± 2.1%) (*F*_2,72_ = 17.208, *p* < 0.001). The intraspecific competition model (Equation (3)) effectively captured the consumption rate across various ladybird densities, irrespective of the species (Table 3). The asymptotic 95% confidence intervals for the maximum consumption rate (*q*) and the parameter of intraspecific competition (*m*) of all three ladybird species did not include zero. The maximum consumption rate of *H. axyridis* (*q* = 60.9%) or *C. septempunctata* (*q* = 56.6%) was nearly twice that of *H. variegata* (*q* = 35.6%). The impact of intraspecific competition on *H. variegata* (*m* = 0.41) was more pronounced compared to *H. axyridis* (*m* = 0.33) and *C. septempunctata* (*m* = 0.35).

### 3.3. Semi-Field Study 

The optimal prey-to-predator ratios for ladybirds feeding on aphids were determined to be 231.8 (*H. axyridis*), 284.2 (*C. septempunctata*), and 180.6 (*H. variegata*), based on the theoretical maximum consumption (*N*_max_) calculated using Equation (2). The utilization of ladybirds in practical applications was demonstrated as an effective approach to managing aphid populations (*F*_3,8_ = 38.772, *p* < 0.001; Figure 3). In the absence of ladybirds, the aphid population increased by 53.6% ± 16.4% in 30 days. In contrast, *H. axyridis* (83.9% ± 5.2%) and *C. septempunctata* (78.7% ± 5.0%) exhibited a more pronounced reduction in the aphid abundance compared to *H. variegata* (27.3% ± 9.9%) within a span of 30 days.

## 4. Discussion

Our study demonstrated that ladybirds should be recognized as natural enemies of aphids and potentially valuable agents in biological control programs by decision-makers in goji berry orchards. Both our laboratory and field studies provided evidence of the ability of ladybirds to suppress aphids on goji berry plants. The pest pressure of aphids was not exceptionally high during our study. However, the abundance of aphids per foliage reached a maximum average of 9.2 in the ladybird-free plots of the most-infested orchard. This level of infestation would have exceeded the tolerance threshold for farmers. The plots introduced with ladybirds consistently exhibited an average of less than two aphids, a level that would be deemed acceptable by most standards. Our findings support previous studies indicating the significant role of ladybirds as effective predators of aphid species [29]. Additionally, our results emphasize the potential underestimation of ladybirds by decision-makers in goji berry orchards, who could benefit from considering their inclusion in integrated pest management strategies.

We observed a type II functional response against aphids by all adult ladybirds of three species. Relevant studies have also documented type II functional response curves in ladybird predation on diverse prey species [35,36,37,38,39]. Holling’s disc equation is widely used as the predominant model for analyzing the type II functional response [40,41]. The results of our study also revealed that, for three species of ladybirds, the consumption of aphids by ladybirds increased proportionally with the number of aphids provided. However, once a certain threshold was reached, the consumption gradually slowed down and remained below this threshold level. This pattern can be accurately described using Holling’s disc model. These findings were consistent with previous research, indicating that Holling’s disc model is an appropriate framework for describing the ladybird consumption of various prey species [42,43,44,45].

The mean aphid consumption by *H. axyridis* and *C. septempunctata* exceeded that of *H. variegata*, potentially attributed to the relatively smaller size of *H. variegata*. However, the DNA-based analysis of gut contents suggests that *H. variegata* exhibits a higher predation rate on *A. gossypii* compared to *C. septempunctata* (90.6% vs. 70.9%) in melon crops [23], which can be attributed to factors such as the ladybug’s digestive capacity and feeding timing, potentially resulting in contrasting outcomes. Additionally, the reduced predation by *H. variegata* on goji berry plants may be attributed to the disparate nutritional value of *A. gossypii* when feeding on goji berries as opposed to melons [29]. The efficiency of *H. axyridis* in locating aphids was found to be the highest, potentially attributed to the invasive nature of *H. axyridis*, which allows for greater adaptability and more flexibility [46]. Predators exhibit exceptional sensitivity to the direction and angle of prey, leading them to efficiently locate their target [47,48]. In this regard, *H. axyridis* demonstrates rapid recognition of prey location based on reference objects in its surroundings. The mobility of *H. axyridis* facilitates its detection, thereby augmenting its searching efficiency on aphids. The handling time required for *C. septempunctata* to process one aphid was the shortest, possibly due to its higher nutritional requirements for reproduction [49]. The larger size of *C. septempunctata* may indicate a higher level of aggression and voracity compared to the other two species, and these characteristics may also result in a potentially greater maximum theoretical consumption by *C. septempunctata.*

The findings of our study revealed that, despite maintaining a constant aphid-to-ladybird ratio of 100, the consumption rate of ladybirds exhibited a progressive decline with increasing ladybird density in the Petri dish. The consumption rate of ladybirds was negatively affected at high ladybird densities due to an increased likelihood of intraspecific competition or mutual interference resulting from resource or space limitations [50]. The intraspecific competition model for ladybirds has been proven to yield the parameters *q* (maximum consumption rate) and *m* (parameter of intraspecific competition) [51,52].

The maximum consumption rate of *H. axyridis* or *C. septempunctata* was found to be twice as high as that of *H. variegata*, while the influence of intraspecific competition on *H. variegata* exceeded that on *H. axyridis* or *C. septempunctata*. However, the presence of *H. variegata* is likely to be further reduced due to the hypothetical artificial increase of both *H. axyridis* and *C. septempunctata*, as they are asymmetric intraguild predators of *H. variegata* [23,53]. The consumption rate of ladybirds, being digestive-limited predators, exhibits a limitation due to predator satiation [54]. The activity of the ladybirds increased significantly following the consumption of a large number of aphids, which likely promoted intraspecific competition. Therefore, it is possible that a considerable amount of time was dedicated to intraspecific competition when the ladybirds had reached a low level of satiation [55,56]. Therefore, the stronger intraspecific competition among *H. variegata* individuals may be attributed to their lower maximum consumption rate, given their small size. This reduction may be insignificant for aphid control if the impact of *H. variegata* is low, but this may not be the case considering that this species is recognized for its biocontrol potential in the goji berry field. If the efficacy of larvae is indeed confirmed, one could consider releasing *H. variegata* as a less aggressive intraguild predator. Alternatively, the introduction of *H. axyridis* or *C. septempuctata* could be proposed, but strategies for mitigating the risk of intraguild predation on *H. variegata* should also be taken into account.

In the semi-field study, all three species of ladybirds demonstrated high efficacy in controlling aphid populations. The absence of ladybirds on plants resulted in a 53% increase in aphid populations within 30 days, whereas *H. axyridis*, *C. septempunctata*, and *H. variegata* exhibited the potential to cause reductions ranging from 27.3% to 83.9%. Comparatively, the biological control efficiency of *H. axyridis* and *C. septempunctata* against the aphids was superior to that of *H. variegata*, indicating that *H. axyridis* and *C. septempunctata* may be more suitable choices for controlling aphids in the goji berry field. However, the suitability of *H. variegata* as a predator for aphids can still be observed under specific circumstances [57]. Significantly, the field release of ladybird eggs has demonstrated cost-effectiveness by obviating the necessity for larval mass rearing and providing convenience in terms of their transportation, storage, and portability [58].

The estimation of a predator’s potential as a biological control agent relies heavily on consumption traits such as searching efficiency and handling time [59], and prey–predator dynamics can be assessed through the utilization of a mathematical model [60]. Thus, the aphid–ladybird dynamics can be established by analyzing the functional response curves, and the consumption ability of ladybirds is contingent upon the aphid density in their natural habitats. However, the validation of aphid–ladybird dynamics requires field studies, as quantitative models developed in laboratory or semi-field studies seem to have limited applicability in assessing foraging abilities under actual field conditions, likely due to variations in predator search efficiency under different conditions [61,62]. The spatial complexity, which plays a crucial role in the natural environment, cannot be replicated under simplified semi-field or laboratory conditions. These laboratory studies provide a parametric analysis of predator-dependent intraspecific competition models, but they are limited to a non-spatial scale. Therefore, it is crucial for future research to consider intraspecific competition in a spatial context as it is closer to natural conditions [63,64,65]. The presence of intraspecific competition has the potential to disrupt the foraging capacities quantified by a functional response. Therefore, the comprehension of not only prey–predator but also predator–predator interactions is crucial for a reliable aphid control strategy based on ladybirds. 

The semi-field study further confirmed the significant potential of *H. axyridis* and *C. septempunctata* as promising candidates for aphid control in goji berry orchards, but the selection between them may also depend on the specific ladybird’s environmental adaptability. Therefore, it is crucial to choose the dominant ladybird species based on the prevailing environmental conditions. However, *H. variegata* also exhibits potential as a biological control agent due to its enhanced fitness, enabling it to effectively respond to fluctuations in pest density. The availability of pollen and nectar from flowers as supplementary food sources in the marginal vegetation of crops can enhance the reproductive performance of *H. variegata*, particularly when prey is scarce [66]. The indoor feeding of ladybirds on a large scale is indispensable if they are to be released into fields in the future. Therefore, disparities in feeding efficiency between naturally occurring and laboratory-reared ladybirds must also be considered, given that the ladybirds utilized in this study were all collected from the field. Moreover, the influence of natural stochasticity and intraguild predation should be taken into account in future field experiments investigating intraspecific competition among ladybirds [67,68]. The consideration of plant characteristics is also essential due to their influence on the feeding efficiency of ladybirds [69]. Future field studies associated with prey–predator dynamics are crucial for the successful implementation of ladybird release, as it has the potential to minimize pesticide usage and enhance the predator population [70].

## 5. Conclusions

The aphid *A. gossypii* is a significant pest that inflicts severe damage upon goji berry plants in China. To assess the potential of ladybirds for aphid biological control, we conducted laboratory experiments to evaluate the functional responses of three ladybird species: *H. axyridis*, *C. septempunctata*, and *H. variegata.* Additionally, we examined intraspecific competition among ladybird individuals. The practical efficacy of ladybirds in reducing aphid populations was investigated under semi-field conditions. A functional response model accurately described the foraging behaviors of ladybirds, while an intraspecific competition model indicated the presence of mutual interference among ladybird individuals that impacted their prey consumption. Therefore, conducting a comprehensive analysis of the functional response and intraspecific competition will contribute to enhancing our understanding of prey–predator and predator–predator interactions in the context of biological control against aphids on goji berry plants. The functional response study suggested the efficacy of ladybirds as efficient biological control agents for managing aphids, and the semi-field study further confirmed the significant potential of *H. axyridis* and *C. septempunctata* as promising candidates. The consideration of an appropriate density is also crucial to mitigate intraspecific competition when releasing ladybirds in the field, as an insufficient or excessive number of released ladybirds may compromise the effectiveness of the biological control program. However, future field studies are crucial in ensuring the effective implementation of a biological control program.

## Figures and Tables

**Figure 1 insects-14-00853-f001:**
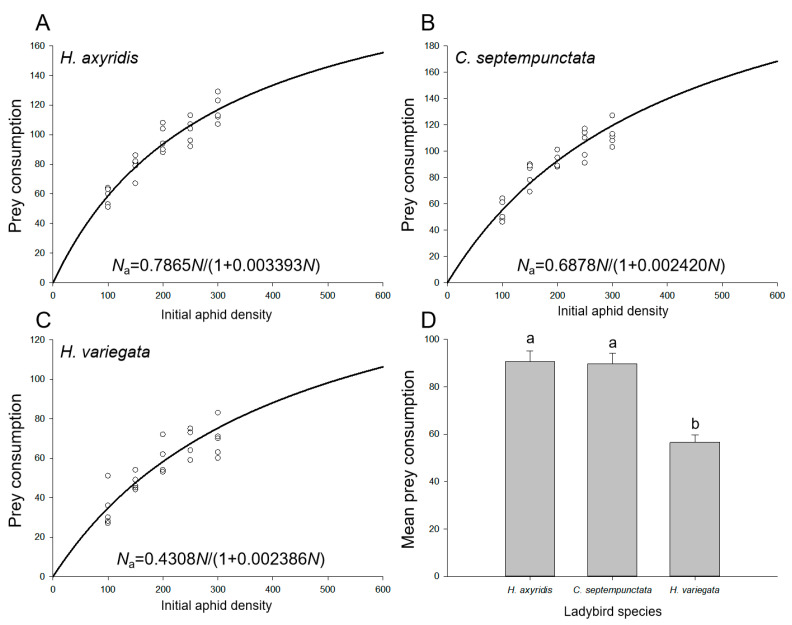
Ladybird consumption of aphids on goji berry plants. Functional response of (**A**) *H. axyridis*, (**B**) *C. septempunctata*, and (**C**) *H. variegata*. Solid lines show the functional response curves of ladybirds attacking aphids obtained by fitting Holling’s disc model (Equation (2)). Circles indicate the number of aphids consumed at each aphid density. (**D**) Mean consumption by the three ladybird species. Different letters indicate significant differences among the ladybird species (mean separation by Tukey’s HSD, *p* < 0.05).

**Figure 2 insects-14-00853-f002:**
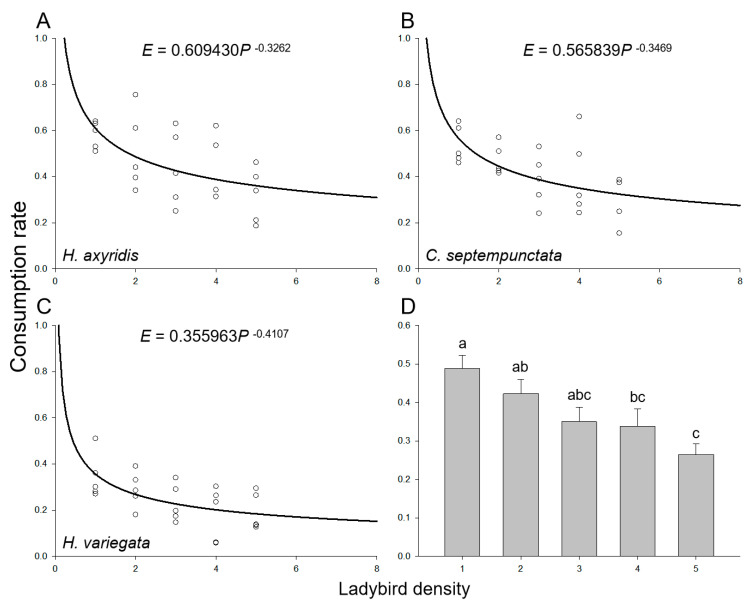
Consumption rate of ladybirds preying on aphids on goji berry plants. Intraspecific competition among adult individuals of (**A**) *H. axyridis*, (**B**) *C. septempunctata*, and (**C**) *H. variegata*. Circles represent the mean consumption rate per ladybird at each ladybird density. The curve was fit using the intraspecific competition model (Equation (3)). (**D**) Mean consumption rate at various ladybird densities. Different letters indicate significant differences among treatments (mean separation by Tukey’s HSD, *p* < 0.05).

**Figure 3 insects-14-00853-f003:**
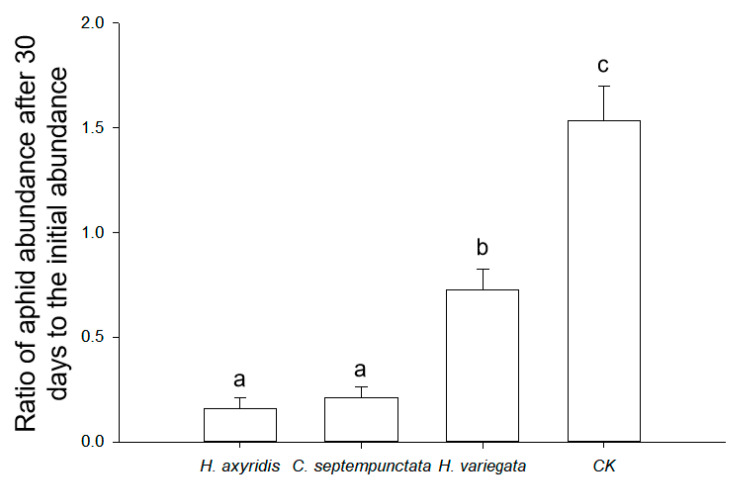
Ratio of aphid abundance after 30 days to the initial abundance in semi-field conditions. The treatments include the presence of *H. axyridis*, *C. septempunctata*, or *H. variegata*, and the absence of ladybirds (CK). Different letters indicate significant differences among treatments (mean separation by Tukey’s HSD, *p* < 0.05).

**Table 1 insects-14-00853-t001:** Maximum likelihood estimates (±SE) for parameters of the logistic model fit to the proportion of aphids consumed versus the initial aphid density.

Stage	*P* _0_	*P* _1_	*P* _2_	*P* _3_
*H. axyridis*	2.470 **	−0.0512 **	4.1 × 10^−4^	−9.7 × 10^−7^
(SE)	0.354	0.0270	7.0 × 10^−5^	1.5 × 10^−7^
*C. septempunctata*	2.183 *	−0.0041 *	−3.6 × 10^−5^	9.1 × 10^−8^
(SE)	0.546	0.0360	8.0 × 10^−6^	2.0 × 10^−8^
*H. variegata*	−0.215	−0.0072 *	4.0 × 10^−5^	−6.3 × 10^−8^
(SE)	0.094	0.0060	7.0 × 10^−6^	9.0 × 10^−9^

* Significant at *p* < 0.05; ** Significant at *p* < 0.01.

**Table 2 insects-14-00853-t002:** Parameter estimates of Holling’s disc equation for ladybirds preying on aphids on goji berry plants.

Ladybird Species	*R* ^2^	*F*	*P*	*a*	*T*_*h*_ (min)	*N* _max_
*H. axyridis*	0.994	478.234	<0.001	0.79	6.21	231.8
*C. septempunctata*	0.97	98.655	0.002	0.69	5.07	284.2
*H. variegata*	0.982	161.44	0.001	0.43	7.97	180.6

*R*^2^ is the coefficient of determination estimated by fitting Holling’s disc equations; *P* is the probability that Holling’s disc equation yields parameters; *a* is the searching efficiency; *T*_*h*_ is the handling time; *N*max is the theoretical maximum consumption.

**Table 3 insects-14-00853-t003:** Parameter estimates of the intraspecific competition equation of the consumption rate of ladybirds at various densities.

Ladybird Species	*R* ^2^	*F*	*P*	*q*	*m*
*H. axyridis*	0.849	16.88	0.026	60.9%	0.33
*C. septempunctata*	0.821	13.723	0.034	56.6%	0.35
*H. variegata*	0.94	47.05	0.006	35.6%	0.41

*R*^2^ is the coefficient of determination estimated by fitting intraspecific competition equations; *P* is the probability that intraspecific competition equation yields parameters; *q* is the maximum consumption rate; *m* is the intraspecific competition parameter.

## Data Availability

The data presented in this study are available on request from the corresponding author.

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
