# Peer review of "Functional Response and Intraspecific Competition of Three Ladybird Species Feeding on Aphids on Goji Berry Plants in Laboratory and Semi-Field Conditions"

_insects, 2023, doi:10.3390/insects14110853_

Round 1

Reviewer 1 Report

Comments and Suggestions for Authors

The authors present a study on functional response and intraspecific competition of three ladybird species feeding on A. gossypi. The results are interesting and the methodology used is adequate and is well explained in the material and methods section. The article is worth to be published but there are some aspects that should be taken into account to improve the publication.

General comments

1) The title is too uninformative and does not conform to the usual standards of a scientific publication. In addition the keywords do not cover the absence of scientific names in the title which makes it difficult to search in databases. It is not indicated which species of aphid is being studied, nor what exactly is 'goji berry plant' (in the title), nor is it indicated what is meant by 'ladybird consumption'.

A more precise title would be 'Functional response and intraspecific competition of three species of ladybird predators of the aphid Aphis gossypii Glover on the goji berry plant Lycium barbatum L. in laboratory and semi-field conditions' or similar, although some scientific/common names could be moved to keywords.

2) The english is sometimes confusing (see examples in particular comments). This is specially notorius in the first sentence of discussion which is literally wrong: should be something like "Our study showed that ladybirds should be considered important biological control agents by decision-makers in goji berry orchards". I would recommend replace in the proposed sentence 'biological control agents' by 'natural enemies' and add 'and potentially a valuable agent in biological control programs' (to differentiate between 'natural control' and 'biological control').

3) The discussion is well structured but some notes on penultimate and last paragraphs.

The penultimate paragraph on semi-field study: it is not necessary to repeat the same data with their errors that have already been given in results. As in previous paragraphs discuss the data not to repeat them. Also, it is surprising that there is no reference in the discussion to this data, e.g. Int. J. Entomol. Res. 03 (03) 2015 on functional response of H. variegata. The authors should seek further reference for discussion of this data.

The last paragraph of discussion is excessively long and deals with marginal topics of the study. It should be rewritten and simplified to focus in the need of studies in field condition and the differences/limitations of the laboratory and semi-field studies.

In relation with the discussion is surprising that nothing is pointed about an fundamental aspect of functional response (and biological control using ladybirds) as is the supplementary food (see e.g. J. Econ. Entomol. 101(6): 1792-1797 -2008-) on H. variegata. It is a key topics that the authors would be discuss both the importance and because was not included in the experiments.

4) The conclusions are not conclusions but seem more like a summary or synthesis of the paper. It should be rewritten to focus in real conclusions and implications of the results; perhaps also future research.

Particular comments

1) The wording should be more in line with scientific terminology, e.g.,

L47: delete 'scientifically identified as'
L55: 'Pest control on' instead 'Refraining from'
L68: 'honeydew' instead 'aphid secretion'
L81: 'natural enemies' instead only 'predators' (the context conservation and management is general to speak only of predators)
L244: 'ranging from' when is an enumeration of values not range
L250: 'approximattely twice' is very imprecise (61/35 = 1.7; 57/35=1.6; not 2.0), 2.3 is also approx. twice. Better e.g. almost twice or near twice
L321: Aphis citricola not A. citricol (missing last 'a')
and so on

2) Statistical analysis

* the authors use some equations without references to the source. Add references of the equations 1, 2, and 3
* the authors use t-test, ANOVA, Tukey HSD test on count data. The authors should demonstrate that the data are normally distributed, count data usually follow a Poisson distribution not a Normal distribution. If so P-values are not well estimated.

3) Tables

* Table 1: fewer decimal places in P0 and use negative powers of 10 to avoid so many zeros in P1, P2, P3
* Table 2: it is better to put the equations in the corresponding subfigures of figure 1, not in the table

4) Figures

* Figure 2: it is missing to indicate which part of the caption corresponds to the subfigure D (I think before 'Mean consumption)
* Figure 3: really is a 'ratio of... to initial abundance'?. What mean a ratio of 0 as basis? usually the basis of a ratio is 1.
I think this figure is unnecessary, the data is in the text with more precision or put a short label Y 'percentage of population change' in relation to the text data.

5) References
Missing some references of functional responses e.g.

* Journal of Insect Science 5(1) 2005: Functional responses of coccinellid predators: al least to discuss because the authors not follow this kind of analysis
* or the previouly pointed: International Journal of Entomology Research or Journal of Economic Entomology
* Also remove references from simplification of the last paragraph of discussion.

Author Response

The authors present a study on functional response and intraspecific competition of three ladybird species feeding on A. gossypi. The results are interesting and the methodology used is adequate and is well explained in the material and methods section. The article is worth to be published but there are some aspects that should be taken into account to improve the publication.

-We are extremely grateful to the reviewer’s comments on our manuscript. We have studied the valuable comments carefully and tried our best to revise the manuscript. Based on the suggestions, we have answered and revised the questions in detail one by one.

General comments

1) The title is too uninformative and does not conform to the usual standards of a scientific publication. In addition the keywords do not cover the absence of scientific names in the title which makes it difficult to search in databases. It is not indicated which species of aphid is being studied, nor what exactly is 'goji berry plant' (in the title), nor is it indicated what is meant by 'ladybird consumption'.

A more precise title would be 'Functional response and intraspecific competition of three species of ladybird predators of the aphid Aphis gossypii Glover on the goji berry plant Lycium barbatum L. in laboratory and semi-field conditions' or similar, although some scientific/common names could be moved to keywords.

-The title has been changed to “Functional response and intraspecific competition of three ladybird species feeding on the aphid on goji berry plants in laboratory and semi-field conditions”.

2) The english is sometimes confusing (see examples in particular comments). This is specially notorius in the first sentence of discussion which is literally wrong: should be something like "Our study showed that ladybirds should be considered important biological control agents by decision-makers in goji berry orchards". I would recommend replace in the proposed sentence 'biological control agents' by 'natural enemies' and add 'and potentially a valuable agent in biological control programs' (to differentiate between 'natural control' and 'biological control').

-Modified “Our study demonstrated that ladybirds should be recognized as natural enemies and potentially valuable agents in biological control programs by decision-makers in goji berry orchards.”

3) The discussion is well structured but some notes on penultimate and last paragraphs.

The penultimate paragraph on semi-field study: it is not necessary to repeat the same data with their errors that have already been given in results. As in previous paragraphs discuss the data not to repeat them. Also, it is surprising that there is no reference in the discussion to this data, e.g. Int. J. Entomol. Res. 03 (03) 2015 on functional response of H. variegata. The authors should seek further reference for discussion of this data.

-The absence of ladybirds on plants resulted in a 53% increase in aphid populations within 30 days, whereas H. axyridis, C. septempunctata and H. variegata exhibited the potential to cause reductions ranging from 27.3% to 83.9%.

-We assume that the reviewer refers to this reference: the reviewer Alizamani, T.; Razmjou, J.; Naseri, B.; Hassanpour, M.; Asadi, A.; Kerr, C. Effect of vermicompost on life history of Hippodamia variegata preying on Aphis gossypii Glover. J Entomol. Res. Soc. 2017, 19, 51-60. We have added the relevant discussion and cited the corresponding references “However, the suitability of H. variegata as a predator for aphids can still be observed under specific circumstances [53].”

The last paragraph of discussion is excessively long and deals with marginal topics of the study. It should be rewritten and simplified to focus in the need of studies in field condition and the differences/limitations of the laboratory and semi-field studies.

-We have significantly simplified the paragraph and divided it into two separate paragraphs. One paragraph specifically addresses the disparity between the laboratory findings and field results, while the other focuses on potential limitations of the experiment.

In relation with the discussion is surprising that nothing is pointed about an fundamental aspect of functional response (and biological control using ladybirds) as is the supplementary food on H. variegata (see e.g. J. Econ. Entomol. 101(6): 1792-1797 -2008-). It is a key topics that the authors would be discuss both the importance and because was not included in the experiments.

-We discussed the topic and cited the literature “In fact, H. variegata exhibits the potential to function as an efficient biological control agent due to its enhanced fitness, enabling it to effectively respond to fluctuations in pest density. The availability of pollen and nectar from flowers as supplementary food sources in the marginal vegetation of crops can enhance the reproductive performance of H. variegata, particularly when prey is scarce [71].”

4) The conclusions are not conclusions but seem more like a summary or synthesis of the paper. It should be rewritten to focus in real conclusions and implications of the results; perhaps also future research.

-In this section, we have made revisions by omitting an overview of the results and incorporating a detailed analysis of the impact and significance of the findings, along with providing insights into future prospects.

Particular comments

1) The wording should be more in line with scientific terminology, e.g.,

L47: delete 'scientifically identified as'

-Modified.

L55: 'Pest control on' instead 'Refraining from'

-Modified.

L68: 'honeydew' instead 'aphid secretion'

-Modified.

L81: 'natural enemies' instead only 'predators' (the context conservation and management is general to speak only of predators)

-Modified.

L244: 'ranging from' when is an enumeration of values not range

-Modified.

L250: 'approximattely twice' is very imprecise (61/35 = 1.7; 57/35=1.6; not 2.0), 2.3 is also approx. twice. Better e.g. almost twice or near twice

-Modified.

L321: Aphis citricola not A. citricol (missing last 'a')

and so on

2) Statistical analysis

* the authors use some equations without references to the source. Add references of the equations 1, 2, and 3

-We have added references 27 to 29 to equations 1 to 3, respectively

* the authors use t-test, ANOVA, Tukey HSD test on count data. The authors should demonstrate that the data are normally distributed, count data usually follow a Poisson distribution not a Normal distribution. If so P-values are not well estimated.

-All data were checked for normal distribution and homoscedasticity before tests, and we added the statement in the Statistical analysis.

3) Tables

* Table 1: fewer decimal places in P0 and use negative powers of 10 to avoid so many zeros in P1, P2, P3

-Modified. P2 and P3 were shown in scientific notation.

* Table 2: it is better to put the equations in the corresponding subfigures of figure 1, not in the table.

-The equations originally presented in Tables 2 and 3 have been relocated to figures 1 and 2, correspondingly.

4) Figures

* Figure 2: it is missing to indicate which part of the caption corresponds to the subfigure D (I think before 'Mean consumption)

-The serial number of sub-figure D was added.

* Figure 3: really is a 'ratio of... to initial abundance'?. What mean a ratio of 0 as basis? usually the basis of a ratio is 1.

I think this figure is unnecessary, the data is in the text with more precision or put a short label Y 'percentage of population change' in relation to the text data.

-The presentation of Figure 3 has been revised to enhance the clarity and comprehensibility for readers, illustrating the aphid abundance ratio before and after a period of 30 days.

5) References

Missing some references of functional responses e.g.

* Journal of Insect Science 5(1) 2005: Functional responses of coccinellid predators: at least to discuss because the authors not follow this kind of analysis

* or the previouly pointed: International Journal of Entomology Research or Journal of Economic Entomology

-The aforementioned three references have been incorporated into the discussion section, where the functional response of ladybugs is examined in light of these sources. “Relevant studies have also documented type II functional response curves in ladybird predation on diverse prey species. [32-36]”

* Also remove references from simplification of the last paragraph of discussion.

-The reference has been removed.

Reviewer 2 Report

Comments and Suggestions for Authors

General comments:

Wu et al. compared the consumption response, intraspecific competition and population control effect of three ladybird species for control to Aphis gossypii, a serious pest in goji berry production in China. Overall, this is a fairly straight paper. The flow and methodology of the study has been presented clearly and the data are logically analyzed and stated. This manuscript provides some evidence for further field application of ladybird in management of aphid pest in goji orchard. 

Specific suggestions to improve the manuscript are as follows.

Specific comments:

The author should improve the English writing of the manuscript. For example, in the first line of abstract, “Aphis gossypii Glover, is identified as a significant pest”. In my opinion, “significant” is not suitable here, should be changed to “serious”.

line 284-285, the authors propose to use ladybirds in field. Actually, except the control effect, cost is a critical factor. So, I recommend the authors could add some related discussion, which would provide more evident for the use of ladybirds in goji production.

line 305-309, the authors compared consumption among the three ladybirds, however, which one would be more suitable to be applied for aphid control in goji orchard might also depend on the environmental adaptability of the specific ladybird. Which ladybird is the dominant species following aphid occurrence should be discussed, especially under the specific environmental conditions. This would be also very help information guiding the use of ladybirds for control of A. gossypii in goji orchard.

Comments on the Quality of English Language

The author should improve the English writing of the manuscript.

Author Response

General comments:

Wu et al. compared the consumption response, intraspecific competition and population control effect of three ladybird species for control to Aphis gossypii, a serious pest in goji berry production in China. Overall, this is a fairly straight paper. The flow and methodology of the study has been presented clearly and the data are logically analyzed and stated. This manuscript provides some evidence for further field application of ladybird in management of aphid pest in goji orchard.

-We are extremely grateful to the reviewer’s comments on our manuscript. We have studied the valuable comments carefully and tried our best to revise the manuscript. Based on the suggestions, we have answered and revised the questions in detail one by one.

Specific suggestions to improve the manuscript are as follows.

Specific comments:

The author should improve the English writing of the manuscript. For example, in the first line of abstract, “Aphis gossypii Glover, is identified as a significant pest”. In my opinion, “significant” is not suitable here, should be changed to “serious”.

-We have changed the text to "serious" and checked the full text for English. 

line 284-285, the authors propose to use ladybirds in field. Actually, except the control effect, cost is a critical factor. So, I recommend the authors could add some related discussion, which would provide more evident for the use of ladybirds in goji production.

-We improved the discussion of the cost of natural enemy release. “Significantly, the field release of ladybird eggs has demonstrated cost-effectiveness by obviating the necessity for larval mass rearing and providing convenience in terms of their transportation, storage, and portability.” 

line 305-309, the authors compared consumption among the three ladybirds, however, which one would be more suitable to be applied for aphid control in goji orchard might also depend on the environmental adaptability of the specific ladybird. Which ladybird is the dominant species following aphid occurrence should be discussed, especially under the specific environmental conditions. This would be also very help information guiding the use of ladybirds for control of A. gossypii in goji orchard.

-We added the environmental adaptability of the specific ladybird to the discussion based on reviewer’s comment. “The semi-field study further confirms the significant potential of H. axyridis and C. septempunctata as promising candidates for aphid control in goji berry orchards, but the selection between them may also depend on the specific ladybird's environmental adaptability. Therefore, it is crucial to choose the dominant ladybird species based on the prevailing environmental conditions. However, H. variegata also exhibits the potential as biological control agent due to its enhanced fitness, enabling it to effectively respond to fluctuations in pest density. The availability of pollen and nectar from flowers as supplementary food sources in the marginal vegetation of crops can enhance the reproductive performance of H. variegata, particularly when prey is scarce [70].”

Reviewer 3 Report

Comments and Suggestions for Authors

In this manuscripts authors have evaluated in laboratory and semi-field conditions the functional response of three species of ladybirds, Harmonia axyridis, Coccinella septempunctata, and Hippodamia variegata, on the aphid Aphis gossypii infesting goji plants. Curiously they noticed that H. axyridis and C. septempunctata have higher impact on the aphid. Previous studies on the functional response of C. septempunctata and H. variegata larvae on A. gossypii have been conducted in laboratory (https://academic.oup.com/ee/article-abstract/28/2/307/486425, https://academic.oup.com/ee/article/32/1/151/490302, https://www.mdpi.com/2075-4450/5/4/974, and https://link.springer.com/article/10.1007/s10340-011-0387-9) or in melon field using molecular gut content analysis of predation (https://www.nature.com/articles/s41598-018-20830-2 ). In this respect, the sentence in L83-84 is not correct, as potential for biological control of A. gossypii by the three ladybird species has been raised multiple times. I strongly encourage authors to refer to the abovementioned articles and discuss their results in relation to previous findings.

For instance, in this study by Wu et al., H. variegata did not exhibit an impressive predation on A. gossypii in goji. Conversely in melon crop, larvae of H. variegata were shown to highly predate on A. gossypii (90.6%), at a much higher level than that of C. septempunctata  (70.9%) https://www.nature.com/articles/s41598-018-20830-2 ). This reduced predation by H. variegata could be a consequence of using adults rather then larvae, which are known to be less voracious, and, as correctly written in L300-301, also much smaller than the other two other ladybird species. But more explaination are indeed possible. For instance, could this reduced predation by H. variegata, be a consequence of the different nutritional value of A. gossypii reared on goji, rather than on melon (see https://link.springer.com/article/10.1007/s10340-009-0272-y ) ?

L88: change "Variegata" with "variegata"

L103-105: if I am not wrong, new emerged individuals were not fed until the bioassay. In this very short timeframe one may expect that individuals were not in their reproductive status. Do you think that having reproductive and mated individuals would have changed their feeding behaviour? Was the sex-ratio of tested ladybirds measured?

L187: not just the natural mortality but also the production of newborn individuals.

Table 1: it would be valuable to mention in the materials and methods the meaning of the four parameters P1, P2, P3, and P4 (e.g., plateau, slope, etc.).

L247: table 3 is missing.

L248-29: abbreviations of the function parameters should be reported in materials and methods

L249-252: however, according to what is written in L240-242, this difference is not significant, right?

L335-338 and L395-396: This should be better elaborated. Intraspecific competition has a higher impact on H. variegata rather than in H. axyridis and C. septempunctata (L291-392). However, H. axyridis and C. septempunctata are both asymmetric intraguild predators of H. variegata (https://link.springer.com/article/10.1007/s10526-012-9470-2, https://www.nature.com/articles/s41598-018-20830-2 ) So the hypotetical artificial increase of these two species is likely to further reduce the presence of H. variegata. For aphid control, perhaps this reduction would be negligible if the impact of H. variegata were low, but this could be not the case as this species is recognised as having biocontrol potential in goji (L332-333). If larvae were indeed effective (?), one could consider releasing H. variegata (less aggressive intraguild predator). Alternatively, the release of H. axyridis or C. septempuctata could be suggested, but, at the same time, strategies for reducing the risk of intraguild predation on H. variegata should be also considered.

Author Response

In this manuscripts authors have evaluated in laboratory and semi-field conditions the functional response of three species of ladybirds, Harmonia axyridis, Coccinella septempunctata, and Hippodamia variegata, on the aphid Aphis gossypii infesting goji plants. Curiously they noticed that H. axyridis and C. septempunctata have higher impact on the aphid. Previous studies on the functional response of C. septempunctata and H. variegata larvae on A. gossypii have been conducted in laboratory (https://academic.oup.com/ee/article-abstract/28/2/307/486425, https://academic.oup.com/ee/article/32/1/151/490302, https://www.mdpi.com/2075-4450/5/4/974, and https://link.springer.com/article/10.1007/s10340-011-0387-9) or in melon field using molecular gut content analysis of predation (https://www.nature.com/articles/s41598-018-20830-2 ). In this respect, the sentence in L83-84 is not correct, as potential for biological control of A. gossypii by the three ladybird species has been raised multiple times. I strongly encourage authors to refer to the abovementioned articles and discuss their results in relation to previous findings.

-Thank you very much for the reviewer's comments. When we discussed this problem at that time, we always discussed it as "wolfberry (goji berry) aphids" rather than "cotton aphids". Since wolfberry aphids have only recently been identified as mainly cotton aphids, our previous studies have been conducted as wolfberry aphids, and the literature found is all about "wolfberry aphids", so we have ignored the relevant literature on finding functional responses in cotton aphids. We have revised the statement that previous studies focused primarily on the functional response of ladybug larvae to cotton aphids in the laboratory and in melon fields; our research focuses on the functional response of adult ladybugs to cotton aphids on goji berry plants. The relevant literature is also added.

-“Regarding the predator, functional responses of ladybird larvae on A. gossypii have been investigated both in laboratory settings and in the melon field using molecular gut content analysis to assess predation [19-23]. However, there is currently limited research on the potential of adult ladybirds in controlling the aphid on goji berry plants.”

For instance, in this study by Wu et al., H. variegata did not exhibit an impressive predation on A. gossypii in goji. Conversely in melon crop, larvae of H. variegata were shown to highly predate on A. gossypii (90.6%), at a much higher level than that of C. septempunctata (70.9%) https://www.nature.com/articles/s41598-018-20830-2 ). This reduced predation by H. variegata could be a consequence of using adults rather then larvae, which are known to be less voracious, and, as correctly written in L300-301, also much smaller than the other two other ladybird species. But more explaination are indeed possible. For instance, could this reduced predation by H. variegata, be a consequence of the different nutritional value of A. gossypii reared on goji, rather than on melon (see https://link.springer.com/article/10.1007/s10340-009-0272-y ) ?

-Thanks to the reviewer for a good discussion point. Predation criteria in the literature provided by the reviewer may be different from ours. The predation of H. variegata larva here is higher than that of C. septempunctata larva (90.6% vs. 70.9%, not predation efficiency here), just based on the results of detectability of prey DNA. The results of DNA detectability can be influenced by factors such as the digestive capacity of the ladybug's gut and the timing of feeding (which happens to be the time when H. variegata eats more but has less digestive capacity, resulting in the detection of more prey DNA).

-As far as we know, H. variegata is much smaller than C. septempunctata, both adult and larva. It is difficult for the consumption of H. variegata to exceed that of C. septempunctata. In addition, the predation tests in the literature also suggest that the consumption of C. septempunctata is higher than that of H. variegata “IGP was similar among the two species, although corrected data might suggest a stronger predation by Cseptempunctata.”

-We have added relevant expressions to the discussion: “The mean aphid consumption by H. axyridis and C. septempunctata exceeded that of H. variegata, potentially attributed to the relatively smaller size of H. variegata. However, the DNA-based analysis of gut contents suggests that H. variegata exhibited a higher predation rate on A. gossypii compared to C. septempunctata (90.6% vs 70.9%) in melon crops [23], which can be attributed to factors such as the ladybug's digestive capacity and feeding timing, potentially resulting in contrasting outcomes. Additionally, the reduced predation by H. variegata on goji berry plants may be attributed to the disparate nutritional value of A. gossypii when feeding on goji berries as opposed to melons [43]”.

L88: change "Variegata" with "variegata"

-Modified.

L103-105: if I am not wrong, new emerged individuals were not fed until the bioassay. In this very short timeframe one may expect that individuals were not in their reproductive status. Do you think that having reproductive and mated individuals would have changed their feeding behaviour? Was the sex-ratio of tested ladybirds measured?

-We guess the reviewer refers to this sentence: “Prior to the experiments, ladybirds were transferred into Petri dishes (9 cm diameter), containing goji berry leaves infested with aphids”. We may not have made it clear here that the ladybugs are the colonies of ladybugs used in the experiment, not the specific individual ladybugs used in the experiment. The main description is the feeding process of this generation of ladybirds used in the experiment before its adult stage. The individuals we selected before the experiment were all adult ladybugs within 12 hours of pupation, so there was no feeding and mating. We revised the expression to make it clearer: “The ladybird colonies selected were transferred into Petri dishes (9 cm diameter) containing goji berry leaves infested with aphids”.

L187: not just the natural mortality but also the production of newborn individuals.

-Right, we have indicated that in the text: “The control treatments were conducted to account for the natural mortality of adult aphids and the number of newborn nymphs”.

Table 1: it would be valuable to mention in the materials and methods the meaning of the four parameters P1, P2, P3, and P4 (e.g., plateau, slope, etc.).

-The plateau, direction and slope of the cubic equation curve are determined by the four parameters together, and each parameter does not have a specific curve significance like the quadratic equation, so it is difficult to accurately express it in the text.

L247: table 3 is missing.

-Added.

L248-29: abbreviations of the function parameters should be reported in materials and methods

-The parameters of the functional response are indicated not only in the Statistical analysis of the material and method, but also in the preceding sentence of the expression: “The numerical values of the parameters for search efficiency (a) and handling time (Th) demonstrated this relationship, with asymptotic 95% confidence intervals excluding zero, across three ladybird species.”

L249-252: however, according to what is written in L240-242, this difference is not significant, right?

-Table 1 is the expression explaining this part, which is significant: “The estimated values of the linear parameter P1 were found to be significantly negative for all three species of ladybirds”.

L335-338 and L395-396: This should be better elaborated. Intraspecific competition has a higher impact on H. variegata rather than in H. axyridis and C. septempunctata (L291-392). However, H. axyridis and C. septempunctata are both asymmetric intraguild predators of H. variegata (https://link.springer.com/article/10.1007/s10526-012-9470-2, https://www.nature.com/articles/s41598-018-20830-2 ) So the hypotetical artificial increase of these two species is likely to further reduce the presence of H. variegata. For aphid control, perhaps this reduction would be negligible if the impact of H. variegata were low, but this could be not the case as this species is recognised as having biocontrol potential in goji (L332-333). If larvae were indeed effective (?), one could consider releasing H. variegata (less aggressive intraguild predator). Alternatively, the release of H. axyridis or C. septempuctata could be suggested, but, at the same time, strategies for reducing the risk of intraguild predation on H. variegata should be also considered.

-According to the reviewer's suggestion, we have added a discussion in this section: “The maximum consumption rate of H. axyridis or C. septempunctata was found to be twice as high as that of H. variegata, while the influence of intraspecific competition on H. variegata exceeded that on H. axyridis or C. septempunctata. However, the presence of H. variegata is likely to be further reduced due to the hypothetical artificial increase of both H. axyridis and C. septempunctata, as they are asymmetric intraguild predators of H. variegata [23,55]. ” and “Therefore, the stronger intraspecific competition among H. variegata individuals may be attributed to their lower maximum consumption rate, given its small size. The reduction may be insignificant for aphid control if the impact of H. variegata is low, but this may not be the case considering that this species is recognized for its biocontrol potential in the goji berry field. If the efficacy of larvae is indeed confirmed, one could consider releasing H. variegata as a less aggressive intraguild predator. Alternatively, the introduction of H. axyridis or C. septempuctata could be proposed, but strategies for mitigating the risk of intraguild predation on H. variegata should also be taken into account.”

Round 2

Reviewer 3 Report

Comments and Suggestions for Authors

In this revised version Authors have sensibly improved the manuscript. Only one aspect remains to be clarified.

In L259-261 of the revised version is mentioned that However, irrespective of the ladybird species, the consumption rate of ladybirds decreased with higher densities of both aphids and ladybirds due to intraspecific competition resulting from limited space (Fig. 2A-C)“. And this is also clearly supported by statistics in L263 and Fig 2D. However the abovementioned sentence may suggest that there is no difference in consumption rate among the three ladybird species. Was this difference statistically evaluated ?

If the difference among the three species has not been evaluated, or if that was not significant, the sentence in L270-271 The impact of intraspecific competition on H. variegata (m = 0.41) was more pronounced compared to H. axyridis (m = 0.33) and C. septempunctata (m = 0.35) must be lowered down has there is no statistical support for this difference in intraspecific competition among H. variegata and C. septempunctata or H. axyridis. Also the sentence in the discussion (L355-356) must be reworded accordingly.  

After clarification of this aspect, I think that the manuscript can be accepted for publication.

Author Response

In this revised version Authors have sensibly improved the manuscript. Only one aspect remains to be clarified.

In L259-261 of the revised version is mentioned that “However, irrespective of the ladybird species, the consumption rate of ladybirds decreased with higher densities of both aphids and ladybirds due to intraspecific competition resulting from limited space (Fig. 2A-C)“. And this is also clearly supported by statistics in L263 and Fig 2D. However the abovementioned sentence may suggest that there is no difference in consumption rate among the three ladybird species. Was this difference statistically evaluated ?

-Thanks very much to the reviewer's suggestion, we have added the comparison of average consumption of the three ladybug species under intraspecific competition: “Besides, the mean consumption rates of H. axyridis (45.5%±3.1%) and C. septempunctata (41.5%±2.7%) were higher than that of H. variegata (24.7%±2.1%) (F2,72 = 17.208, P < 0.001)”.

If the difference among the three species has not been evaluated, or if that was not significant, the sentence in L270-271“The impact of intraspecific competition on H. variegata (m = 0.41) was more pronounced compared to H. axyridis (m = 0.33) and C. septempunctata (m = 0.35)“ must be lowered down has there is no statistical support for this difference in intraspecific competition among H. variegata and C. septempunctata or H. axyridis. Also the sentence in the discussion (L355-356) must be reworded accordingly.  

-Indeed, it is necessary to state that the consumption rate of H. axyridis and C. septempunctata is significantly higher than that of H. variegata, so that the expression of L270 is reasonable.

After clarification of this aspect, I think that the manuscript can be accepted for publication.